# Nanoalignment by critical Casimir torques

Gan Wang[1,9], Piotr Nowakowski [2,3,4,9], Nima Farahmand Bafi[2,3,5],
Benjamin Midtvedt[1], Falko Schmidt[6], Agnese Callegari [1], Ruggero Verre[7],
Mikael Käll [7], S. Dietrich[2,3], Svyatoslav Kondrat [2,3,5,8] ✉ & Giovanni Volpe [1] ✉

The manipulation of microscopic objects requires precise and controllable forces and torques. Recent advances have led to the use of critical Casimir forces as a powerful tool, which can be finely tuned through the temperature of the environment and the chemical properties of the involved objects. For example, these forces have been used to self-organize ensembles of particles and to counteract stiction caused by Casimir-Liftshitz forces. However, until now, the potential of critical Casimir torques has been largely unexplored. Here, we demonstrate that critical Casimir torques can efficiently control the alignment of microscopic objects on nanopatterned substrates. We show experimentally and corroborate with theoretical calculations and Monte Carlo simulations that circular patterns on a substrate can stabilize the position and orientation of microscopic disks. By making the patterns elliptical, such microdisks can be subject to a torque which flips them upright while simultaneously allowing for more accurate control of the microdisk position. More complex patterns can selectively trap 2D-chiral particles and generate particle motion similar to non-equilibrium Brownian ratchets. These findings provide new opportunities for nanotechnological applications requiring precise positioning and orientation of microscopic objects.

The manipulation of microscopic objects, such as colloids and nanoparticles, is essential in various research fields, including nanotechnology[1–3] and materials science[4–8]. However, controlling these objects can be challenging due to their small size and to the presence of Brownian motion. To overcome these challenges, one often utilizes methods requiring the use of external fields, such as optical[9–12] and magnetic tweezers[13–16], in order to control the motion of microparticles. However, these methods have limitations in terms of precision and scalability, which impede their application when accurate placement, manipulation, and alignment are required in the near field.

Recently, critical Casimir forces have emerged as a powerful tool to control the motion of micro and nanoparticles[17,18]. These forces, which are the thermodynamic analog of quantum-electro-dynamical (QED) Casimir forces, act on neighboring objects in a critical fluid and can be finely tuned via the temperature of the environment[19,20], the composition of the fluid[21,22], and the chemical properties of the involved objects[23,24]. Importantly, critical Casimir forces can be attractive or repulsive depending on the adsorption preferences of the involved surfaces (e.g., hydrophilic or hydrophobic surfaces)[19,25]. The tunability of these forces has been exploited to control the motion of microscopic particles, achieving trapping[26], bonding[27,28], synchronization[29,30], and even the assembly of particles into micro- and nanostructures[31–35]. Some studies have also shown the potential of patterned substrates to control the motion of microparticles[18,36,37].

[1]Department of Physics, University of Gothenburg, SE-41296 Gothenburg, Sweden. [2]Max Planck Institute for Intelligent Systems, Heisenbergstraße 3, D-70569 Stuttgart, Germany. [3]IV th Institute for Theoretical Physics, University of Stuttgart, Pfaffenwaldring 57, D-70569 Stuttgart, Germany. [4]Group of Computational Life Sciences, Division of Physical Chemistry, Ruđer Bošković Institute, Bijenička cesta 54, 10000 Zagreb, Croatia. [5]Institute of Physical Chemistry, Polish Academy of Sciences, 01-224 Warsaw, Poland. [6]Nanophotonic Systems Laboratory, Department of Mechanical and Process Enginnering, ETH Zürich, CH-8092 Zürich, Switzerland. [7]Department of Physics, Chalmers University of Technology, SE-41296 Gothenburg, Sweden. [8]Institute for Computational Physics, University of Stuttgart, Allmandring 3, D-70569 Stuttgart, Germany. [9]These authors contributed equally: Gan Wang, Piotr Nowakowski. ✉e-mail: svyatoslav.kondrat@gmail.com; giovanni.volpe@physics.gu.se

While the use of Casimir forces has developed into a well-established research field, the potential of QED and critical Casimir torques remains largely unexplored. Indeed, QED torques have been demonstrated experimentally only recently[38,39], while critical Casimir torques have mainly been investigated theoretically[40–43]. For instance, Ref. 40 used mean-field theory to study ellipsoidal particles at a flat homogeneous wall; in refs. 41,43 critical Casimir torques have been simulated in two spatial dimensions; and in ref. 44 torques driven by depletion interactions have been investigated theoretically. More recently, in ref. 42, forces and torques between two patchy particles have been studied numerically using the Derjaguin approximation. Their approximate results agree with experiments reported in ref. 45, which examined the formation of switchable structures with patchy particles. However, the use of critical Casimir torques to localize, align, and manipulate the orientation of particles is still to be established.

In this study, we demonstrate that critical Casimir torques provide a powerful tool to control the nanoscopic alignment of microscopic objects on nanopatterned substrates. We experimentally show that circular patterns can stabilize the vertical position and orientation of nanofabricated disks (silica (SiO$_2$), radius of ca. 1 µm, and thickness of ca. 400nm) immersed in a critical binary liquid mixture (water–2,6-lutidine). Using the Derjaguin approximation, we theoretically show how a delicate balance of critical Casimir repulsion and attraction from different substrate regions can localize a microdisk and induce its vertical alignment. Furthermore, we experimentally demonstrate how more complex patterns – such as elliptical, triangular, and spiral

patterns – can enhance microdisk trapping, selectively trap chirally-symmetric particles, and even propel particles along critical Casimir ratchets. These findings open the door for accurate manipulation and alignment of microscopic objects, covering nanotechnological applications which range from particle sorting and separation to optomechanics and nanomachinery.

## Results

### Particle trapping at nanopatterned substrates

We consider the trapping of a spherical and a disk-shaped microparticle suspended in a water–2,6-lutidine critical mixture (the critical lutidine concentration amounts to the mass fraction 0.286, and the lower critical temperature is $T_c \approx 310K \approx 34\,°C$; see Methods "Critical mixture") above a patterned substrate, as illustrated in Fig. 1a. The substrate consists of a 25nm-thick patterned gold film deposited on a fused silica (SiO$_2$) substrate. Circular openings with diameters between 1 µm and 2.8 µm were obtained by a combination of electron beam lithography (EBL), evaporation and lift-off process (see Methods "Substrate fabrication" and Supplementary Fig. S1); a scanning electron microscope (SEM) image of this substrate is shown in Fig. 1b. In order to control the wetting properties of this substrate, we chemically functionalized the patterned gold film with hydrophobic thiols[46] and made the SiO$_2$ circular patterns hydrophilic by applying an oxygen plasma (see Methods "Substrate fabrication" and Supplementary Fig. S1).

We synthesized microspheres with a diameter of $(3 \pm 0.1)$ µm using the hydrothermal method[47]. We fabricated the microdisks

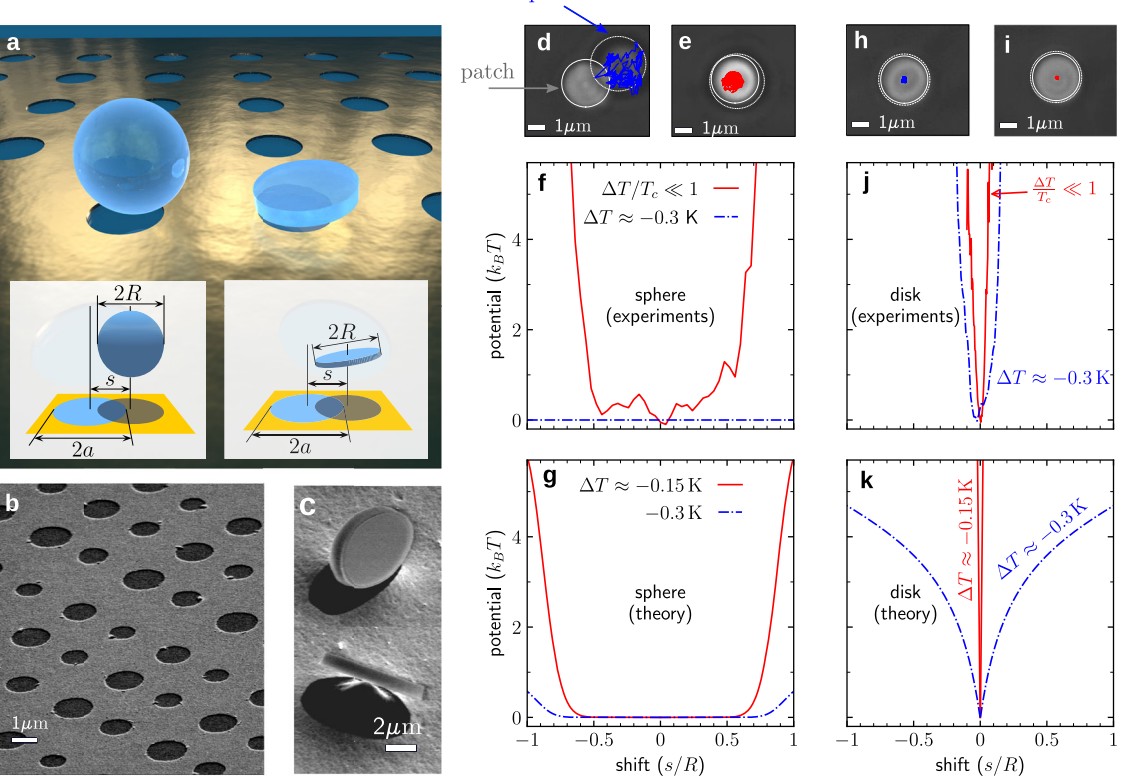

**Fig. 1 | Trapping of microparticles above a nanopatterned surface. a** Artist rendition of a spherical (left) and a disk-shaped (right) microparticle trapped above a circular uncoated pattern within a thin gold layer coated on a glass surface. The insets illustrate the notation used in this article. **b** Scanning electron microscope (SEM) images of a gold-coated glass surface with circular patterns of diameters 2*a* between 1 µm and 2.8 µm. The thickness of the gold-coating is 25nm. **c** SEM images of microdisks (upright coins) with diameter 2*R* = 2.4 µm. **d** A microsphere is freely diffusing in the *xy*-plane (blue trajectory) at $\Delta T = (-0.30 \pm 0.02)$K off the critical temperature $T_c$, **e** while it gets confined above the circular pattern (red trajectory)

for $\Delta T = (0.00 \pm 0.02)$K, i.e., much closer to $T_c$. **f** Experimentally measured potentials at $\Delta T = (-0.30 \pm 0.02)$K (blue lines) and $\Delta T = (0.00 \pm 0.02)$K (red lines). **g** Theoretically predicted potentials at $\Delta T \approx -0.3$K (blue lines) and $\Delta T \approx -0.15$K (red lines). **h** A microdisk is trapped already at $\Delta T = (-0.30 \pm 0.02)$K (blue trajectories), and (**i**) even more strongly at $\Delta T = (0.00 \pm 0.02)$K (red trajectories). This is confirmed by (**j**) the experimentally measured and (**k**) theoretically calculated potentials. See Supplementary Fig. S8 for calculations of forces and torques acting on the microdisk. Source data are provided as a Source Data file.

(diameter (2.4 ± 0.1) μm and thickness 400 nm) by patterning a thermally oxidized silicon wafer and releasing the structure using a combination of laser lithography and reactive ion etching (see Methods "Microdisk fabrication" and Supplementary Fig. S2). An exemplary SEM image is shown in Fig. 1c. Both the microspheres and microdisks are hydrophilic, being made of $SiO_2$[48].

Depending on the wetting properties of the substrate and the particles, either attractive or repulsive critical Casimir forces can arise. Attractive critical Casimir forces emerge in the presence of similar wetting properties (e.g., hydrophilic particles above a hydrophilic $SiO_2$ substrate), while repulsive critical Casimir forces emerge in the presence of opposite wetting properties (e.g., hydrophilic particles above a hydrophobic gold substrate), as the temperature $T$ of the sample approaches $T_c$. We used a two-stage feedback temperature controller, which stabilized the temperature of the sample with a precision of ± 0.02K[18,20,49] (see Methods "Experimental setup" and Supplementary Fig. S3).

Up to a temperature about 0.30 K below $T_c$, the microsphere diffused freely above the substrate (Supplementary Video 1). A typical trajectory is shown in blue in Fig. 1d. All the trajectories were tracked with DeepTrack 2, which provides a deep learning framework to track particle positions with high accuracy[50] (see details in Methods "Particle detection and tracking"). As the temperature was raised towards the lower critical point at $T_c$, an attractive critical Casimir force emerged between the $SiO_2$ microsphere and the $SiO_2$ pattern as well as a repulsive critical Casimir force between the microsphere and the gold substrate. Thus, the microsphere was trapped near the center of the pattern, as shown by the red trajectory in Fig. 1e. In order to quantify the trapping of the particle, we used the particle trajectories to compute an effective confining potential $U_{exp}(s)$. This potential was calculated by evaluating the probability $P_{exp}(s)$ for the particle displacement from the center of the pattern ($s = 0$) and by using $U_{exp}(s) = -k_B T \ln P_{exp}(s)$. Figure 1f (red line) shows that at $T \approx T_c$, the microsphere trapping is due to a rectangular flat-bottom potential over the pattern, while there is no confinement if $\Delta T = T - T_c \approx -0.30K$ (blue line). These experimental results agree well with the corresponding theoretical predictions shown in Fig. 1g (see Methods "Model interaction potential").

By repeating the experiment with a microdisk, we observed stable trapping already at a temperature about 0.50K below $T_c$, i.e., the microdisk underwent a transition from free motion above the substrate to confinement at the pattern further away from $T_c$ compared to the microsphere. The blue and red lines in Fig. 1h, i are the trajectories at $\Delta T = -0.30K$ and at $T \approx T_c$, respectively, showing that the microdisk is confined in both cases. Similarly as for the microsphere, from the experimentally measured trajectories we determined effective confining potentials $P_{exp}(s)$. The main features of $P_{exp}(s)$ agree well with the theoretical calculations (see Methods "Model interaction potential"), as Fig. 1j, k demonstrate.

Interestingly, the experimentally observed lateral trapping of the microdisk is much more effective than that of the microsphere. Indeed, for the microsphere we observed no trapping at $\Delta T \approx -0.30K$, while for $|\Delta T|/T_c \ll 1$ we found $\sigma_x = 305nm$ for the displacement standard deviation in the $x$-direction (Supplementary Fig. S4). In contrast, for the microdisk, we obtained $\sigma_x = 55nm$ at $\Delta T \approx -0.30K$ and $\sigma_x = 22nm$ at $|\Delta T|/T_c \ll 1$. This difference arises as a consequence of the different effective interaction areas. Due to the limited extent of critical fluctuations at $T \neq T_c$, only the bottom part of the microsphere (closest to the substrate) effectively interacts with the substrate. In contrast, for a microdisk oriented parallel to the substrate, the area of its interaction with the substrate is much larger, spanning essentially the whole microdisk area. Furthermore, while the microsphere diffuses over the area of the $SiO_2$ pattern, as demonstrated by the flat-bottom potential in Fig. 1f, g the microdisk is strongly confined above the center of the pattern. The microdisk confinement arises due to the

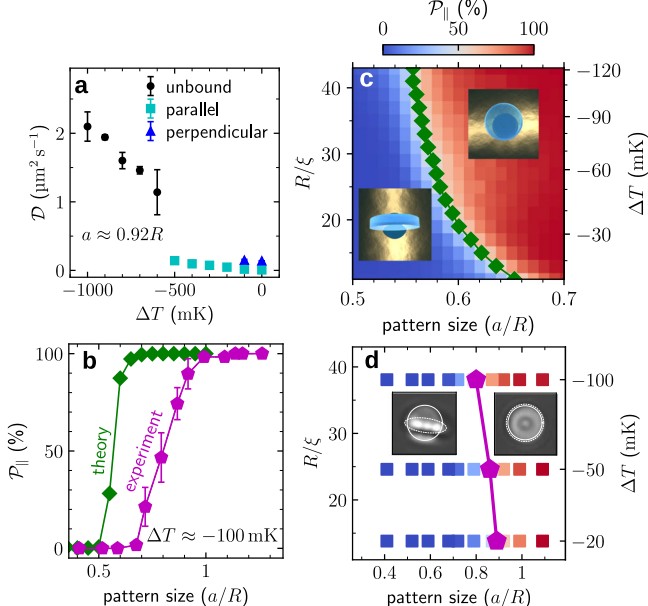

**Fig. 2 | Microdisk alignment by critical Casimir forces. a** Diffusion coefficient of a microdisk (radius $R = 1.2μm$) above a circular pattern (radius $a = 0.92R$) as a function of temperature $\Delta T = T - T_c$. Far from $T_c$ (black circles), the microdisks are freely diffusing in the fluid. As the temperature rises (cyan squares), the microdisks get trapped above the pattern parallel to the surface. As the temperature rises even further (blue triangles), critical Casimir torques can also flip the microdisks into a configuration perpendicular to the surface. **b** Experimental (purple pentagons) and theoretical (green diamonds) probability of a parallel configuration, $\mathcal{P}_{\parallel}$, of the microdisk above a circular pattern as a function of $a$ at $\Delta T = T - T_c = -100mK$ (see Methods "Measurement of the configuration"). **c** Theoretical $\mathcal{P}_{\parallel}$ as a function of $a$ and $\Delta T$ (right axis) and $R/\xi$ (left axis), where $\xi = \xi_0(\Delta T/T_c)^{-\nu}$ is the fluid correlation length, $\xi_0$ is the bare correlations length, and $\nu \approx 0.63$ is the bulk critical exponent. The green diamonds separate the two phases corresponding to $\mathcal{P}_{\parallel} > 50\%$ (red region) and $\mathcal{P}_{\parallel} < 50\%$ (blue region). The insets schematically illustrate the two configurations. **d** Experimental $\mathcal{P}_{\parallel}$ as a function of $a$ and $\Delta T$. The squares indicate the points for which the experiment was performed, and the purple pentagons locate the boundary between the two phases corresponding to $\mathcal{P}_{\parallel} > 50\%$ (red points) and $\mathcal{P}_{\parallel} < 50\%$ (blue points). The lines are guides for the eye. The insets show the microscope images of the two configurations. Source data are provided as a Source Data file.

presence of a repulsive critical Casimir force between the particle and the gold substrate which surrounds the pattern. When reaching the rim of the pattern, the microdisk is pushed back towards the center, resulting in a stable trapping behavior.

## Microdisk alignment

While so far the orientation of the microdisk has always stayed parallel to the substrate (Fig. 1), one can control also its orientation above the substrate by tuning the temperature. Figure 2a shows the measured diffusion coefficient of the microdisk (radius $R = 1.2μm$) above a pattern of radius $a \approx 1.1μm$ as $T$ is increased towards $T_c$ (see Methods "Measurement of the mean square displacement and of the diffusion constant"). For $\Delta T < -0.50K$ (i.e., far from the critical temperature), the microdisk diffused freely above the substrate; the corresponding diffusion constant $\mathcal{D}$ is large, albeit slowly decreasing from $\mathcal{D} \approx 2 μm^2/s$ to $\mathcal{D} \approx 1 μm^2/s$, as the temperature increases and the microdisk approaches the substrate[51–54] (black circles in Fig. 2a). At $\Delta T \approx -0.50K$, we observe a sharp decrease to $\mathcal{D} = 0.07 μm^2/s$, indicating that the microdisk is close to the substrate (Supplementary Fig. S11), where it is trapped laying flat above the center of the pattern, as shown in Supplementary Video 2. As we increased $T$ further towards $T_c$, the diffusion coefficient slightly decreased to $\mathcal{D} = 0.02 μm^2/s$, indicating a stronger trapping of the microdisk at the substrate (cyan squares in Fig. 2a). For

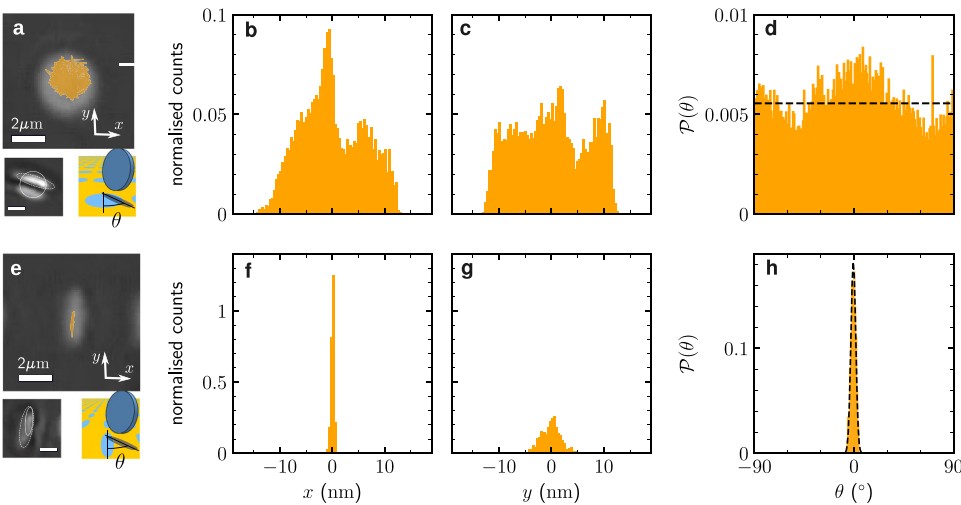

**Fig. 3 | Microdisk position and orientation control by elliptical and circular patterns. a** Trajectory of a microdisk of radius $R = 1.2\mu m$ trapped in the perpendicular configuration above a circular pattern of radius $a = 1.1\mu m$. The bottom panels in (**a**) show a micrograph of the trapped microdisk (left) and its schematic representation (right). The white solid line highlights the circular pattern on the surface, and the white dashed line highlights the microdisk oriented almost perpendicularly to the surface. The microdisk tilting leads to an ellipse-like, rather than a rectangular contour, as seen in the micrograph. The angle $\theta$ within the surface is measured between an arbitrarily chosen axis of the circular pattern (denoted by the short black vertical segment) and the projection of the microdisk diameter chosen to lie parallel to the surface. The shaded area shows the orthogonal projection of the microdisk onto the surface. The micrograph shows the top view while the schematic drawing presents a side view in order to provide a clearer illustration of the angle $\theta$. **b**, **c** Normalized one-dimensional histograms of the microdisk displacements along the $x$- and $y$-directions. **d** The orientation of the microdisk in

(**a**) defined by the angle $\theta$ is nearly uniformly distributed from $-90°$ to $90°$. The dashed line shows the homogeneous probability density $\mathcal{P} = 1/180°$. **e** Trajectory of a microdisk levitating above an elliptical pattern (long axis $2.2\mu m$ and short axis $0.6\mu m$). The bottom panels show the experimental picture (left) and the schematic representation (right) of the trapped microdisk. Similarly to panel (**a**), the solid and dashed lines highlight the contours of the surface pattern and of a circular microdisk oriented perpendicularly to the surface, respectively. The angle $\theta$ is measured as indicated in panel (**a**), taking the long axis of the ellipse as a reference axis. The microdisk displacements along the $x$-direction (**f**) are much more confined than along the $y$-direction (**g**). **h** The orientation of the microdisk is sharply confined between ca. $-7°$ and $7°$. The dashed line shows a Gaussian fit to the experimentallzy measured histograms (orange) with a mean value of $\theta \approx -0.58°$ and a variance of $2.19(°)^2$. All scale bars correspond to $2\mu m$. The temperature is close to $T_c$, i.e., $\Delta T/T_c \ll 1$. The white bars in (**a**) and (**e**) represent $2\mu m$. Source data are provided as a Source Data file.

---

temperatures closer to $T_c$ than $\Delta T \approx -0.10$K, the microdisk started switching between two configurations: either laying flat on the substrate (cyan squares in Fig. 2a) or standing perpendicular to the substrate (blue triangles in Fig. 2a), as shown in Supplementary Video 2.

The coexistence of parallel and perpendicular configurations is a consequence of the delicate balance between the repulsive and attractive critical Casimir forces due to the hydrophilic $SiO_2$ circular pattern and the hydrophobic gold substrate surrounding it. Accordingly, this coexistence depends on the geometrical parameters of the pattern and of the microdisk. We quantified it by measuring the probability $\mathcal{P}_\parallel$ that a microdisk is parallel to the substrate as a function of the ratio $a/R$ of the pattern radius to the microdisk radius. The experimental results (purple pentagons in Fig. 2b) show that there is a transition from the parallel configuration ($\mathcal{P}_\parallel = 100\%$) to a perpendicular configuration ($\mathcal{P}_\parallel = 0\%$) as $a/R$ decreases from 1 to 0.7, i.e., as the size of the pattern decreases, thus increasing the repulsive critical Casimir forces. A qualitatively similar result is observed from the theoretical calculations (green diamonds in Fig. 2b, see Methods "Monte Carlo simulations"), even though the transition occurs at a smaller value of $a/R$.

In order to gain more insight into this phenomenon, we employed Monte Carlo simulations (see Methods "Monte Carlo simulations") to determine a phase diagram in the plane spanned by $a/R$ and $\xi/R$, where $\xi$ is the characteristic length of the critical fluctuations, which can be mapped onto the temperature difference $\Delta T = T - T_c$. The results (Fig. 2c) indicate a sharp transition between the two configurations, which sensitively depends on $\Delta T$ and the pattern size. We obtained similar results from the experiments (Fig. 2d), albeit the location of the transition was slightly shifted, which is likely due to the approximations we have made in modeling the interactions between the

microdisk and the patterned surface (see Methods "Model interaction potential").

## Enhanced microdisk localization and orientation by noncircular patterns

When trapped in the perpendicular configuration above a circular pattern, the microdisk is poorly localized in the xy-plane, as shown by the trajectory in Fig. 3a. Such an unstable trapping is a consequence of the small overlap area between the microdisk and the substrate, resulting in a weak critical Casimir force. Moreover, as a consequence of the rotational symmetry of this system, the orientation of the microdisk freely diffuses featuring a uniform angular distribution, shown in the histogram in Fig. 3d.

To better control both the localization and the orientation of the microdisk, we used an elliptical pattern, instead of a circular one. As shown in Fig. 3e, the microdisk gets trapped above this elliptical pattern in the perpendicular configuration with a much better translational confinement than above the circular one, especially along the short $x$-axis (standard deviations $\sigma_x = 6$nm and $\sigma_y = 37$nm, respectively). Furthermore, the orientation of the microdisk gets pinned along the long $y$-axis of the pattern, resulting in a narrow orientation distribution (Fig. 3h, see Supplementary Video 3). Thus, by patterning the substrate and by tuning the temperature of the environment, we can control the lateral position, the upright or flat configuration, and the orientation (i.e., the angular distribution) of the microdisk with nanometer accuracy.

## Chiral microparticle nanoalignment

Critical Casimir forces provide a powerful tool for the identification, trapping, and manipulation of objects with specific properties on the

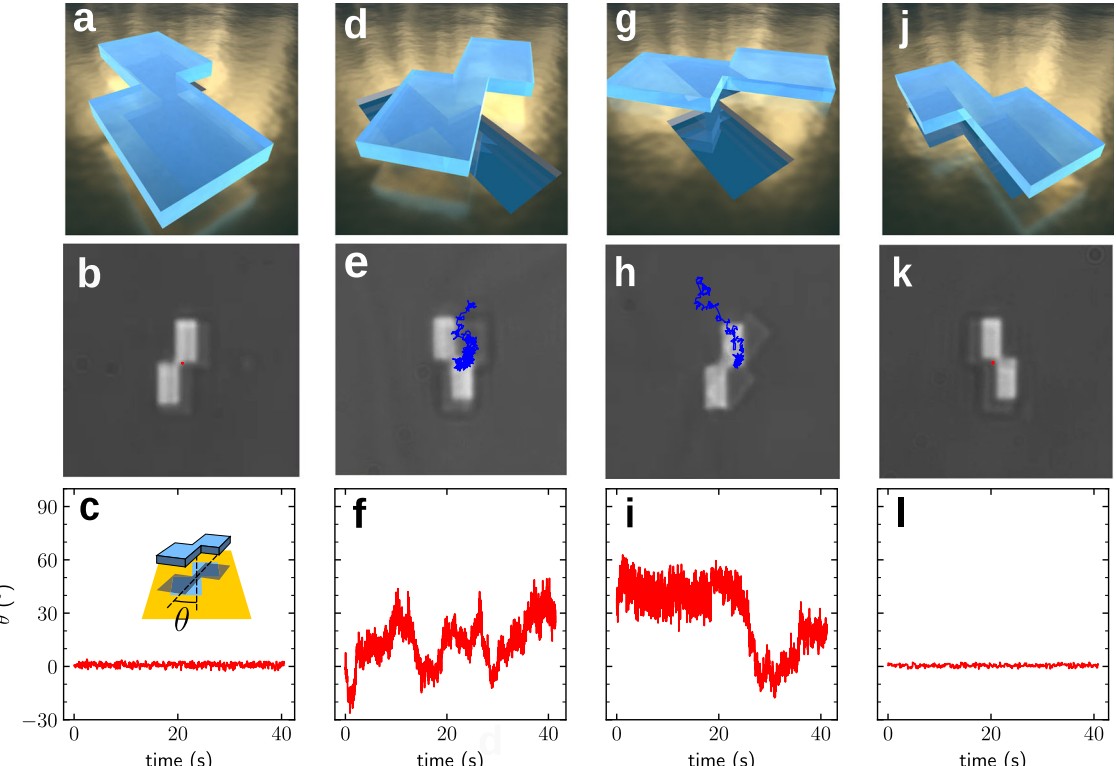

**Fig. 4 | Nanoalignement of chiral microparticles. a** Schematic and **b** brightfield image of a right-handed 2D-chiral microparticle consisting of two overlapping, conjoint together, rectangles with 2.8μm length and 1.8μm height trapped above a right-hand 2D-chiral pattern with the same (slightly smaller) shape and chirality at $T \approx T_c$ (see Supplementary Video 4). The (barely visible) red trajectory shows that the position of the microparticle is well confined at the pattern. **c** Angle $\theta$ between the particle and the pattern (as indicated in the inset) as a function of time showing that the orientation of the particle is locked within a very narrow angular range.
**d**–**f** When the same right-handed 2D-chiral microparticle is trapped above the left-handed pattern, there is less confinement both in position (blue trajectory in **e**) and orientation (panel (**f**)). **g**–**l** Using a left-handed 2D-chiral microparticle leads to similar observations, i.e., it is not strongly confined above a right-handed pattern both in position (**h**) and orientation (**i**), but it is well confined above a pattern with matching (left-handed) 2D-chirality both in position (**k**) and orientation (**l**). Source data are provided as a Source Data file.

micro- and nanoscale. In order to demonstrate that critical Casimir torques provide additional control, we fabricated 2D-chiral particles consisting of two partly overlapping, conjoint rectangles (length 2.8μm and width $w = 1.8$μm), employing the same method used for the fabrication of the microdisks (see Methods "Microdisk fabrication"); the orientation of such 2D-chiral particles can be either right-handed or left-handed. We also fabricated 2D-chiral glass patterns with the same shape but slightly smaller sizes (length 2.5μm and width $w = 1.7$μm). As illustrated in Fig. 4, we considered the behavior of the particles above these patterns in an environment at a near-critical temperature ($T \approx T_c$). A right-handed particle above a right-handed pattern (Fig. 4a) is strongly confined both translationally (as shown by the (barely visible) red trajectory in Fig. 4b) and rotationally (as shown by the particle–pattern angle shown in Fig. 4c). In contrast, when the particle is above a left-handed pattern (Fig. 4d), it is only weakly confined and prone to escape (Fig. 4e); also the rotational confinement is diminished (Fig. 4f). A similar behavior can also be observed for a left-handed particle, which is only weakly confined above a right-handed pattern (Fig. 4g-i), but strongly confined above a left-handed pattern (Fig. 4j-l).

### Critical Casimir ratchet
Finally, we show that substrate micropatterning can be used to control the motion of the microdisks, i.e., that appropriate patterns can prompt the microdisks to move in a specified direction if $T \approx T_c$. We achieved this by fabricating a triangular gold pattern with base 2μm and height 36μm. Figure 5a shows the position of the microdisk (i.e., its center) as a function of time, demonstrating that it moves towards the

base of the triangle. Figure 5b shows trajectories of microdisks above triangles of different heights $h$ but of the same base. In all cases, the microdisk moved towards the base of the triangles.

To gain insight into this process, we calculated the total interaction potential, which comprises the critical Casimir, electrostatic, and gravitational potentials, as a function of the microdisk position above a triangle (see Methods "Energy landscape and motion above triangular patterns"). Figure 5c shows that the interaction potential decreases along the triangle towards its base. This decrease occurs because the area of attraction between the triangle and the microdisk, which possess the same surface properties, increases. Thus, the microdisk moves towards the base where the overlap between the hydrophilic microdisk and the hydrophilic triangle is maximized, which minimizes the interaction potential. In Fig. 5d, we plot the resulting positions of the microdisks calculated along the triangle symmetry axis (i.e., for $x = 0$) as a function of time for triangles with different heights but the same width, demonstrating a similar behavior as in the experiments (Fig. 5b).

These results reveal that adjusting the temperature of the system can control the motion of the microdisk above a triangle. However, within this approach, microdisk transport could only be maintained over short distances and above a single triangle. In order to produce long-ranged transport of microdisks, we built a patterned substrate with a series of trapezoids, each with 18μm height and short and wide bases with widths of 1μm and 2μm, respectively, arranged sequentially on top of each other (Fig. 5e). When the temperature difference was $T - T_c \approx -1.30$K, the microdisk was free to diffuse above the entire substrate, as shown by the initial part of the trajectory (blue line in Fig. 5e) near its starting point (marked by a white cross). When we

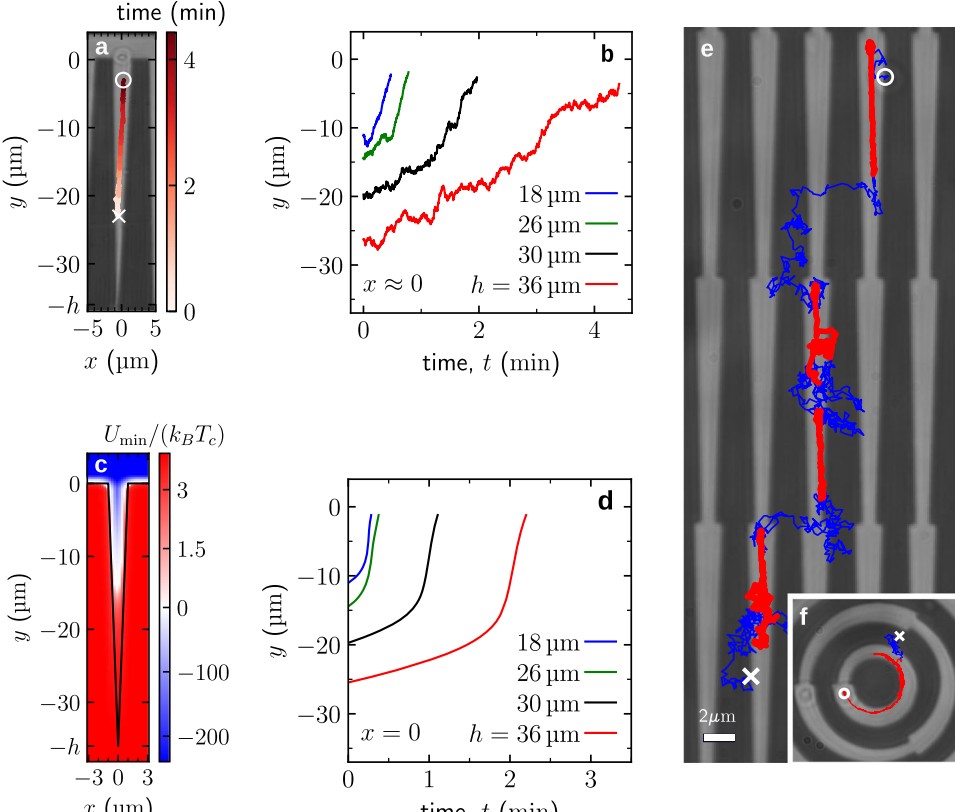

**Fig. 5 | Critical Casimir ratchet. a** A horizontal microdisk of diameter 2.4μm above a triangular pattern with 2μm base and 36μm height at $T \approx T_c$ moves towards the base of the triangle, where the overlap between the microdisk and the growing trapezoid is maximized and, therefore, the critical Casimir potential energy is minimized. **b** Trajectories of the microdisk above triangular patterns with the same base (2μm) but different heights ($h = 18, 26, 30, 36$μm, see Supplementary Video 5). **c** The depth $U_{\min}(x,y)$ of the interaction potential, calculated as a minimum of the potential with respect to the position of a microdisk above a triangular pattern with 2μm base and 36μm height at $T \approx T_c$. **d** Corresponding mean theoretical trajectories above triangular patterns of different sizes, calculated by neglecting Brownian noise. For each $h$, the origin of time is chosen such as to match the position of the microdisk at $t = 0$ with that in the corresponding experiment (see panel (**b**)). The visible speedup is due to steeper changes of the potential close to the base (see panel (**c**)) because the particle gets closer to the surface. **e** Trajectory of a

microdisk above a pattern constituted by a series of trapezoids each with a height of 18μm and short and wide bases of widths 1μm and 2μm. The temperature was cycled so that it was far from critical ($\Delta T = T - T_c \approx -1.30$K) for the blue portions of the trajectory, where the microdisk tends to diffuse freely, and near critical ($T \approx T_c$) for the red portions of the trajectory, where the critical Casimir force pulls the microdisk towards a trapezoid and, subsequently, towards the wide base (see Supplementary Video 6). **f** A similar trajectory above a curved trapezoidal bull-eye pattern, where the microdisk diffuses freely when the temperature is far from critical (blue portion of the trajectory) and follows the bend of the pattern when the temperature is near the critical one (red portion of the trajectory) (see Supplementary Video 7). In panels (**a**), (**e**), and (**f**), the cross and the circle indicate the trajectory starting and finishing points, respectively. Source data are provided as a Source Data file.

increased the temperature to $T \approx T_c$, the microdisk was first trapped above one of the glass trapezoids and then pushed along the trapezoid towards its wide base, as shown by the red portion of the trajectory near the cross in Fig. 5e. By repeating the temperature variation (2 minutes at low temperature and 15 minutes at high temperature, see Methods "Temperature protocol" and Supplementary Fig. S5), the microdisk continued to move along the trapezoids in a ratchet-like manner, thus realizing a critical Casimir ratchet[55]. Figure 5f shows that it is also possible to produce a curved trajectory by bending the trapezoids (see also Supplementary Video 7).

## Discussion

We have demonstrated theoretically and experimentally that critical Casimir forces and critical Casimir torques can controllably align and manipulate spherical and disk-like particles above substrates patterned with nanometer precision. For instance, we switched the microdisk configuration between a parallel and a perpendicular orientation by adjusting the system temperature and the circular pattern radius. Elliptical patterns enhanced the microdisk localization and stabilized its orientation. We also developed

patterned surfaces and critical Casimir forces for selectively trapping chiral particles. Moreover, we showed how to control the linear and circular motion of the particles over long distances (tens to hundreds of micrometers) by combining patterns with continuously varying widths, thus providing the first experimental demonstration of a critical Casimir ratchet.

Our methods introduce a novel way to manipulate particle orientation and motion with high resolution by using critical Casimir torques. This technique will enable future controlled functional assemblies of objects with growth restrictions limited to specific substrates and requiring a secondary transfer process, such as micro-LEDs[56] and two-dimensional materials[57]. Moreover, it presents a viable alternative to assembly methods which require object conductivity[58] or magnetism[6], because it only requires the assembly material to exhibit hydrophilic or hydrophobic properties. With the ability to self-align objects in solution, this method can be applied to separate chiral particles. Our approach also allows one to control the dynamics of the angle between the objects and the substrate, thereby providing a platform to study the physical and chemical properties of materials in terms of their orientational degrees of freedom.

## Methods

### Critical mixture

The microparticles were dispersed in a binary liquid mixture of water and 2,6-lutidine at the critical composition of lutidine $c_{L,c} = 0.286$, which has a critical temperature of $T_c \approx 34\,°C$[59]. The solution was confined in a sample cell formed by a microscopic slide and a cover glass with fabricated patterns.

### Substrate fabrication

The patterns, which were used to control the positioning and movement of the particles, were fabricated on a 22mm × 22mm cover glass with a thickness of 130μm. As illustrated in Supplementary Fig. S1, the fabrication was initiated by spinning coat resist consisting of 200nm LOR 3A (4000rpm for 60s, baking at 200 °C for 5 min) and UVN 2300 (2000rpm for 45s, baking at 100 °C for 1.5 min) on the substrate. A 25nm chromium layer was deposited on the resist to render the sample conductive. Subsequently, an electron-beam lithography step was performed to define the features of the patterns in the positive resist (UVN 2300 was exposed at $10\,\mu C/cm^2$ with a current of 10nA, and developed in developer MF-CD26 for 40s). The pattern was then reversed by lifting off 2nm titanium and 25nm gold in hot acetone at 50 °C for 2 h. In order to make the gold-coated part of the substrate hydrophobic, the sample was immersed in a 1 mmol solution of thiols (1-octanethiol) and ethanol overnight[18]. In this way, a hydrophobic self-assembled layer was formed on top of the gold.

### Microdisk fabrication

The fabrication process of the microdisk is illustrated in Supplementary Fig. S2. The microdisks were fabricated from a 4-inch standard silicon wafer with 400nm thermally grown $SiO_2$. A direct laser writing step was performed in order to fabricate a disk-shaped structure by utilizing a double-layer positive-resist mask (LOR 3A spun at 4000rpm for 60s and baked at 200 °C for 5 min; S1805 spun at 3000rpm for 34s and baked at 110 °C for 1 min). A 40nm hard nickel mask was deposited and lift-off was performed in a subsequent etching process. Then, reactive ion etching was employed to etch $400nmSiO_2$, using 10 sccm (standard cubic centimeters per minute) $CHF_3$ and 15 sccm Ar gas at a pressure of 5 mtorr, with forward and inductively coupled plasma power (FW/ICP) set as 50 W and 600 W, respectively. Following this, a highly selective ion etching with $SF_6$ gas was used in order to etch the Si under the microdisks (50 sccm $SF_3$, 40 mtorr pressure, FW/ICP at 10 W and 300 W, respectively), which left the microdisks to have only small points of contact with the substrate. Finally, the microdisks were sonicated in a binary solution of water−2,6-lutidine at the critical lutidine mass fraction $c_{L,c} = 0.286$.

### Experimental set-up

The experimental set-up is illustrated in Supplementary Fig. S3. Standard digital video microscopy with white light illumination and a CMOS camera was used to capture the motion of the particles. The precise temperature control of the sample was achieved in two stages[18,20,49]. First, the temperature of the sample was kept at (32.5±0.1) °C by a circulating water bath (T100, Grant Instruments), far from the critical temperature $T_c \approx 34$ °C. Second, a feedback controller (Peltier heating/cooling element with a PT100 temperature sensor) was used to control the temperature of the sample with a stability of ± 20mK.

### Particle detection and tracking

The analysis of the particle positions in the video sequences, with the aim to reconstruct the particle trajectories, begins with correcting the drift in the positions of the particles in each frame. This correction is used to eliminate the drift in the images caused by temperature fluctuations during the experiment, which leads to changes in the optical properties of the oil between the sample and the objective. This alignment process hinged on the correlation between each frame and the cumulative average of the preceding frames (for more details, see Supplementary Information Section S1).

After implementing this alignment, the particle localization was performed by using various methodologies depending on the kind of particle. For microdisks trapped perpendicularly at a pattern, we employed a basic binary thresholding method[60], utilizing pixel intensity from the image to discriminate particles from the background and subsequently to extract their contours. The position of the particle was determined by calculating the geometric center of the contours, and the orientation was inferred from the measurement of the major and minor axes of the contours. For microdisks trapped parallelly within a pattern, we employed a convolutional neural network (CNN) trained on synthetic data using the Python package DeepTrack 2[51]. For the microparticles not anchored at a pattern, the LodeSTAR neural network model was utilized, which has the advantage of being a self-supervised method which can be trained on several images of the microdisk configurations without requiring the explicit knowledge of their position[61]. Last, for chiral microparticles located at a pattern, we adopted a hybrid tracking method, starting from manually pinpointing two non-overlapping particle corners to be used for extracting the subsequent particle positions. A more detailed description of the methods and procedures is provided in Supplementary Information Section S1.

In order to ensure the accuracy of the particle position and of the orientation measurements, we employed a covariance-based estimator ref. 62 to calculate the variance of the localization error both for the position and the orientation. The resulting values are 12.8nm and 7.7°, repectively.

### Measurement of the mean square displacement and of the diffusion constant

The mean square displacement (MSD) is defined as

$$\mathrm{MSD}(\tau) = \langle |\mathbf{r}_{t+\tau} - \mathbf{r}_t|^2 \rangle, \tag{1}$$

where $\tau$ is the time interval between the two positions of the particle, $\langle \cdot \rangle$ represents the ensemble average, and $\mathbf{r}_t$ and $\mathbf{r}_{t+\tau}$ are the positions of the particle at times $t$ and $t+\tau$, respectively.

The diffusion constant can be calculated from the variation of the MSD as a function of $\tau$. Specifically, for a freely diffusing particle, the 2-dimensional MSD is expected to increase linearly with $\tau$ for long sequences of positions. The diffusion constant is the corresponding proportionality factor divided by 4. For a constrained or trapped particle, the proportionality is linear only for small $\tau$[60]. In either case, we estimated the MSD from a least squares fit to a linear function for $\tau = \{1, 2, 3, 4\}$ frames, corresponding to a delay of $\approx \{0.033, 0.067, 0.100, 0.133\}$ s.

### Measurement of the configuration

In the microdisk alignment experiment, a diluted solution of microdisks was placed above a nanofabricated substrate with a million circular patterns of different sizes. A considerable number of (ca. 500) of microdisks were immersed in the solution. When the temperature of the sample was increased towards $T_c$, some microdisks were oriented parallel to the patterns, while the remaining ones were standing perpendicularly on top of the patterns. Considering the probability of the parallel configuration as a function of the radius of circular patterns at fixed temperature, we analyzed the probability by counting the number of parallel configurations at a total of 500 patterns of equal size. We repeated this procedure for patterns of various sizes. Regarding the probability of the parallel configuration, as a function of temperature, the counting was performed also on a total of 500 patterns of the same size for three different temperatures.

## Temperature protocol

The periodic modulation of the temperature of the sample towards and away from $T_c$ is shown in Supplementary Fig. S5. An in-house software was used to generate a periodic function that allows one to run the controller unit (TED4015, Thorlabs) in order to create a cyclic temperature change ranging from $T - T_c \approx -1.3\text{K}$ to $T \approx T_c$[18]. The lower temperature was kept for a relatively short time (2 min) in order to prevent the microdisk from diffusing away. Instead, the higher temperature was kept for a longer time (15 min) to allow the microdisk to diffuse above the trapezoidal structure. Through such a temperature cycle, we were able to control the motion of a few micron-sized particles over a long distance on patterned substrates (i.e., hundreds of micrometers).

## Model interaction potential

In all calculations, the interaction potential between a microparticle and a patterned substrate consisted of three contributions:

$$U(\mathcal{P}) = U_c(\mathcal{P}) + U_e(\mathcal{P}) + U_g(D_c), \qquad (2)$$

where $\mathcal{P} = \{D_c, s, \alpha, \gamma\}$ is a set of four parameters defining the microparticle configuration in our set-up. $D_c$ is the distance from the substrate to the center of the microparticle, and $s$ is the lateral shift relative to the center of the circular pattern. The two angles $\alpha$ and $\gamma$ characterize the microdisk orientation and are absent in the case of spherical microparticles. In Eq. (2), $U_c$ is the critical Casimir potential, $U_e$ is the electrostatic potential, and $U_g$ is the gravitational potential.

Following ref. 18, estimates reveal that critical Casimir forces are several times stronger compared to QED Casimir forces (dispersion forces) (Supplementary Information Section S2C). Thus we have neglected such forces in our calculations. In contrast, electrostatic forces are crucial to counterbalance Casimir attraction when particles approach the surface, and gravitational forces are necessary to ensure particle sedimentation at the surface.

Gravitational potential − The gravitational $U_g$ potential can be readily calculated from the gravity acceleration $g$, the volume $v$ and the position $D_c$ of the center of a microparticle, and the difference between the mass densities of the microparticle $\varrho_p$ and of the fluid $\varrho_l$:

$$U_g(D_c) = vg D_c\left(\varrho_p - \varrho_l\right). \qquad (3)$$

The microsphere and microdisk volumes are $v = 4\pi R^3/3$ and $v = \pi R^2 W$, respectively, where $W$ is the disk thickness.

Critical Casimir and electrostatic potentials − We employed the Derjaguin approximation[63] in order to compute the critical Casimir and the electrostatic potential. Within this approximation, an interaction potential is calculated by summing the contributions from thin slices of two interacting objects and taking the limit of infinitesimally small slices, which transforms this sum into an integral. This procedure leads to

$$U_\tau(\mathcal{P}) = \int_A \mathfrak{u}_\tau(\ell(x, y; D_c, \alpha, \gamma))\mathrm{d}x\,\mathrm{d}y, \qquad (4)$$

where $\tau$ element of $\{c, e\}$ denotes the type of interactions (critical Casimir or electrostatic, respectively), and $\mathfrak{u}_\tau(l)$ is the corresponding potential per area calculated for a slab system with two homogeneous, parallel walls separated by a distance $l$. For $\mathfrak{u}_c$, we use formulae for equal or opposite boundary conditions depending on the local chemical properties of the substrate at the position $(x, y)$. The function $\ell(x, y; D_c, \alpha, \gamma)$ is a local distance (measured in the $z$-direction) between the substrate and the surface of the particle. Note that $\ell$ depends on the disk orientation. The set $A$ is an orthogonal projection of the particle onto the substrate plane.

**Critical Casimir interactions.** In the vicinity of a critical point, the critical Casimir interactions are determined by universal scaling functions. For a slab geometry, $\mathfrak{u}_c(l)$ has been obtained by Monte Carlo simulations[64]. In order to simplify the integration in Eq. (4), we used the fitting functions for $\mathfrak{u}_c(l)$ provided in ref. 42.

**Electrostatic interactions** − We adopted the Debye-Hückel approximation to calculate the plate–plate electrostatic potential $\mathfrak{u}_e(l)$. Unlike the scaling function of the critical Casimir potential, which is universal, the Debye-Hückel potential has two free parameters: the Debye screening length $\lambda_D$ and the surface charge density $\sigma$. Since we expect negative charge densities both on hydrophilic and hydrophobic patches of the substrate and on the microdisks, for simplicity we neglected the position dependence of $\sigma$ and assumed it to be the same for the substrate and for the microdisk. While this assumption influences our results quantitatively, we anticipate a qualitatively similar behavior as the role of electrostatic interactions is to counterbalance Casimir attraction for microparticles approaching the substrate. We adjusted the values of $\lambda_D$ and $\sigma$ to qualitatively reproduce the experimental data for the microdisk levitation.

The explicit expressions for $\mathfrak{u}_c(l)$ and $\mathfrak{u}_e(l)$ and examples of the total interaction potential, forces, and torques are provided in Supplementary Information (Section S2 and in Supplementary Figs. S8 and S9).

## Monte Carlo simulations

We employed Monte Carlo (MC) simulations in order to compute the probability $\mathcal{P}_\parallel$ of the parallel configuration shown in Fig. 2b, c. For each set of pattern size and temperature used in Fig. 2b, c, we first identified the two minima in the interaction potential between a disk and a surface, corresponding to the parallel and perpendicular configurations, and estimated an energy barrier $U_b$ between them. The MC moves involved small changes of the parameters $\mathcal{P} = \{D_c, s, \alpha, \gamma\}$ in order to explore the parameter space close to these minima. In addition, we used MC moves for which these parameters were chosen randomly from a homogeneous distribution. This procedure typically resulted in bigger moves, which facilitated switching between the minima.

We performed MC simulations according to the standard Metropolis algorithm[65]. Each simulation consisted of 2000 Monte Carlo moves with 32 small moves per one big move. The final configurations were classified in accordance with the two minima determined above. We considered configurations with energies above $U_b$ as unbound and did not count them to the overall statistics. The probability $\mathcal{P}_\parallel$ was computed as the ratio of the resulting parallel configurations and the total number of bound configurations. In order to gather sufficient statistics, we performed 200 independent runs for each given pattern size and temperature. For further details, see Supplementary Information (Section S4).

## Energy landscape and motion above triangular patterns

Following experimental observations, we assumed that the microdisk is always in the parallel configuration. For simplicity of the calculations, we ignored disk tilting and assumed that the disk is parallel to the substrate. Taking into account that tilt angles in the parallel configuration are small, this assumption influences the interaction potential only slightly. In order to compute the interaction energy $U(x, y, D_c)$ for a microdisk at the position $(x, y, z = D_c)$, we used the interaction potential consisting of the critical Casimir, the electrostatic, and the gravitational interaction energy, similarly as described in the previous subsection "Model interaction potential". Figure 5c shows a heatmap of the potential energy $U_{\min}(x, y) = \min_{D_c} U(x, y, D_c) = U(x, y, D_{c,\min}(x, y))$ computed for the microdisk at the equilibrium distance $D_{c,\min}$ from the substrate with a triangular pattern, corresponding to a local minimum of $U(x, y, D_c)$ with respect to $D_c$.

In order to calculate the trajectory of a microdisk, we assumed an overdamped dynamics suitable for micrometer sized colloidal

particles. Furthermore, we ignored the Brownian motion, i.e., we neglected fluctuations of the disk position and orientation due to thermal motion, thereby focusing on the mean trajectory of the microdisk. These assumptions imply that the friction force counterbalances the force resulting from the interaction with the pattern, leading to the following equation of motion:

$$-\frac{\partial U_{\min}}{\partial y} = \gamma_f \frac{dy}{dt}, \tag{5}$$

where $\gamma_f$ is a friction coefficient, and $y$ is the position of the disk along the triangle main axis. We used the Sutherland-Einstein-Smoluchowski relation and the experimental data for the microdisk diffusion coefficient in order to determine $\gamma_f$. We integrated Eq. (5) numerically in order to find $y(t)$. For further details, see Supplementary Information (Section S5).

## Data availability
The data are available upon request. Source data are provided with this paper.

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

## Acknowledgements

The authors would like to acknowledge funding from the H2020 European Research Council (ERC) Starting Grant ComplexSwimmers (Grant No. 677511) (G.V.), the Horizon Europe ERC Consolidator Grant MAPEI (Grant No. 101001267) (G.V.), and the Knut and Alice Wallenberg Foundation (Grant No. 2019.0079) (G.V.). NFB was supported by the Polish National Science Center (Opus Grant No. 2022/45/B/ST3/00936). We acknowledge scientific discussions with Dr. O. A. Vasilyev regarding this project during his stay at the MPI-IS in Stuttgart. Fabrication in this work was done at Myfab Chalmers.

## Author contributions

S.K. and G.V. designed the research. G.W. performed the experiments and analyzed the experimental data with the help of B.M., F.S., and R.V. P.N. and N.F.B. performed the theoretical calculations and simulations. A.C. performed the QED Casimir calculations. M.K., S.D., S.K., and G.V. supervised the study. All authors contributed to writing and enhancing the paper.

## Funding

## Competing interests

The authors declare no competing interests.
