## [Peer Review File · Nature Communications]

REVIEWER COMMENTS

Reviewer #1 (Remarks to the Author):

This is an interesting work that contributes to the manipulation of microscopic objects by precise and controllable forces and torques. As this study shows, for this purpose critical Casimir forces represent a powerful tool that can be finely tuned through the temperature of the environment and the chemical properties of the involved objects. It is very important to have a tool to self-organize ensembles of particles and to counteract stiction caused by the Casimir-Lifshitz forces.

Here the authors explore for the first time the potential of critical Casimir torques and demonstrate that the critical Casimir torques can efficiently control the alignment of microscopic objects on nanopatterned substrates. They have managed to show experimentally and support with theoretical calculations as well as Monte Carlo simulations that circular patterns on a substrate can stabilize the position and orientation of microscopic disks. And by making the patterns elliptical the microdisks can be subject to a torque which flips them upright while simultaneously allowing for more accurate control of the microdisk position. In addition, as it is elaborated more complex patterns can also selectively trap two dimensional chiral particles and generate particle motion similar to non-equilibrium Brownian ratchets.

All the findings provide new opportunities for nanotechnological applications requiring precise positioning and orientation of microscopic objects, and as a result the publication of this study will be of high interest to several nanotechnology fields.

Therefore, I strongly suggest publication.

Reviewer #2 (Remarks to the Author):

In this manuscript, Gan Wang and co-authors describe their experimental results using critical Casimir forces and torques to align and move floating nanoparticles. By changing the temperature, they can control the strength of the interaction, allowing them to turn the effect on and off. The authors explore the use of disks and spheres for trapping and orientation in- and out-of-plane. They

further describe chiral microparticles and how they align. Finally, the authors describe a critical Casimir ratch, which uses a spatial variation of the hydrophobic surface to create a potential that allows for a ratcheting effect.

Overall the manuscript is well-written, and the results are convincing. I only have a few comments/questions.

1) The focus of the paper is on nanoalignment, but have the authors tried to quantify the forces and torques that they are using? I see the energy potential has been determined but not the force or torque. I'd be curious to see how these numbers compare to the QED Casimir force and torque.

2) Fig. 2 (c,d): η (y-axis) should be defined/clarified. The text mentions that it is a "characteristic length," but I could not find an explicit equation. It is also not mentioned in the caption.

Reviewer #3 (Remarks to the Author):

This article describes a very interesting experiment where critical Casimir force are used to trap particles of different shapes and sizes on a patterned surface. The control parameter is the temperature difference from the critical point, which changes the Casimir interactions between the particles and between the particle and the surface. The very new and interesting result is the appearance of a torque which allows particles to stay in strange equilibrium positions. Thanks to their chirality, they can move when a temperature modulation is applied to the sample. A theoretical interpretation is also proposed. The results are interesting and new and they might have useful applications in particle assembly. For these reasons the article can be published in Nature Comm after that the following comments will be taken into account.

1) The modulation of Casimir force to produce dynamical interactions has been already used in two references which must be quoted : Martinez et al SciPost Phys. 15, 247 (2023) and Martinez et al. Entropy 19, 77 (2017).

2) The discussion of the figure in page 5 and 6 is not well written. Indeed it is written for disks " we determined an effective confining potential $U_{exp}(s)$ by computing the probability $P_{exp}(s)$ " One might think that the method is used for the disks only but I suppose that this is the case for the

measure of the potential acting on the spheres. After the discussion on the disks we go back on the sphere. All of this is very confusing.

3) Page 6. The temperatures are not given both in the text and in the fig.3 caption.

4) Page 10 and methods. The technique to measure theta does not seem very precise because it is based on small light differences. The accuracy must be declared. The accuracy on the position is certainly not great too because the shape of the objects changes a lot when the 3D orientation changes. It would be useful to give an error on the position accuracy.

5) The description of the method used to compute the interactions potential is too short to fully understand the technique. A section has to be added in supplementary materials with more details.

Reviewer #4 (Remarks to the Author):

The manuscript titled 'Nanoalignment by critical Casimir torques' presents a demonstration of the lateral and angular alignment of micron-size objects suspended in a solution above a patterned surface by means of the critical Casimir force. The authors perform tracking experiments to confirm that spherical, disk-shaped and non-rotationally symmetric objects can be guided by shaping an interaction potential and then switching this potential on and off by changing the temperature around the critical point. They also provide numerical computations that show that the most important aspects of the interaction can be understood on the basis of the gravitational, electrostatic and critical Casimir potentials. While these numerical results agree qualitatively with the data, interfacial interactions in liquids are complicated and the assumptions made in the present article may be too crude in some places to achieve quantitative agreement with the data.

While none of the aspects shown in this paper are conceptually new, and many details, such as positional control by surface patterning [1–3], rotation by means of the critical Casimir effect [4], asymmetric structures [4], etc, have been demon-

strated experimentally, it is the first time that the two-dimensional linear and rotational control of single particle trajectories by means of surface patterning and the critical Casimir force are tested explicitly. Also, the flipping of flat disks into the vertical direction is (to my knowledge) new from the experimental perspective. For this reason, the subject is timely and interesting. If the incremental step performed in this work is of sufficient interest for the readers of Nature Communications, is left to be decided by the editor. Apart from this aspect, I recommend publication after minor revision.

General structure and presentation

The paper is written well and clearly understandable. In some parts, such as the description of the theoretical models in Sections S2, and S4–S6, however, the paper lacks detail and many assumptions appear to be ad-hoc to the reader, especially if one is not an expert on the theory in the field of critical Casimir forces (which applies to myself). Here, more in-depth discussions on the taken (simplifying) assumptions, their validity, and impacts on the results would have been beneficial for understanding and clarity. Graphics are mostly well designed and facilitate a better understanding. In some points, however, they appear cluttered and pack too many details, thereby requiring the reader to study the text in all detail to understand the depicted physics. Detailed critiques follow below.

Remarks main text

1. On line 50 it is stated that ‘..., the use of Casimir torques to localize, align, and manipulate the orientation of particles is yet to be established’, which may be a bit exaggerated both in light of the literature review preceding the statement and also in light of existing works, such as Ref. [4] showing the controlled rotation of asymmetric parts both in response to local heating on a

surface and near a wall. Also, the rotational alignment by torques has been demonstrated for the QED Casimir effect [5].

2. Plotting in Figure 2c the phase diagram with coordinate R/ξ instead of ξ/R appears a bit counter-intuitive. Using the inverse would allow the reader to see to directly see the relation of Temperature, particle size and correlation length.

3. Line 180: Should the second width not read σ_y instead of σ_x ?

4. Figure 3: The histograms along the coordinates of the trajectory plot in a and c are so small and without proper scale that they are hard to read and therefore give little information other than the qualitative distribution (which shows that the integration time was probably too short to fully probe the distribution). Also the 'short black vertical segment' mentioned in the caption is very hard to identify. Tablets a and c could be made larger by squeezing the space between the figures. In the caption, you mention that the orientation is 'sharply confined between -7° and $+7^\circ$ '. Does this refer to σ_θ (i.e. the width of a Gaussian distribution), is it the FWHM, or even the min/max of the distribution? A bit more quantitative rigor would be appreciated here.

5. Without reading the supplementary information (and even then) it is very difficult to understand how you modeled and computed the potentials (see comments below).

Remarks on the supplementary information

1. Equation S5 is the basic sum of potentials. Yet, at distances around 100 nm,

despite not being the dominating force, the quantum mechanical Casimir effect should be taken into account, as demonstrated in the literature [3]. On line 366 (methods section of the main article) you note that you follow Hertlein et al (your ref. [19]) to neglect dispersion forces. However, Hertlein et al used a polystyrene particle above a silica surface, which are both dielectric materials leading to strongly reduced QED Casimir forces, while in your case you have SiO₂-Au interactions. Depending on the degree of oxidation of the SiO₂, the original conductivity of the oxidized Si, and the dielectric function of the water-2.6 lutidine mixture, the resulting Casimir forces at ~ 120 nm shortest separations (judging from Figure S7) may be significantly higher than in the experiment of Hertlein et al. It may not reach the strength of the one measured by Schmidt et al [3] who used functionalized gold against a gold flake but also I expect the force in your case not to be lower by an order of magnitude than for Au-Au. Therefore, at least a rough estimate of the expected Casimir potential in relation to the other terms in Equation S5 should be given. This requires knowledge of the dielectric function of the suspension. The respective functions of glass and Au can be taken from the literature.

2. While the use of the Derjaguin approximation should give quantitatively good results from small α , I would expect significant deviations for larger α , γ . This, in addition to the very crude approximation of a homogeneous charge distribution on both the substrate and the microdisk leading to Equation S9, may explain the rather large quantitative differences between the numerical and experimental results in Figure 2 (and Figure 5) of the article. It would be rather simple to replace Equation S9 by an empiric relation or interpolation derived from electrostatic simulations in an appropriate tool (e.g. Comsol or Ansys) taking into account the feedback between $I(x, y)$, U_e (or u_e ?) and

$\sigma(x, y)$ for different α . While this is a nice-to-have feature for the present article in its present state, it would (in my opinion) improve the quantitative understanding of your experiment significantly.

3. Similarly, the assumption that the disk is perfectly parallel to the substrate on line 284 as well as the neglect of 3-body effects in the critical Casimir and electrostatic interactions may imperil quantitative reliability of the numerical results for the ratchets. However, I understand that such computations may be beyond the scope of the present article and given that I am not an expert on the theory you apply, this comment may rather be seen as a curious question.

4. The motivation for the pre-factor of the weight in Equation S10 remains elusive. Due to the central role of this weight, a little discussion would be in order.

5. Distributions for the starting parameters in Monte Carlo simulations are essential for the outcome. Why are uniform distributions for all parameters rectified here? At least for the initial distance to the surface, I would have expected some peaked distribution at the equilibrium distance.

6. The separated single lines visible in Figure S8 (especially b and d) show that the final distribution is probed with insufficient resolution. While the qualitative outcome is clear and also analyzed correctly in the text, the simulation should be repeated until the distribution is sufficiently smooth. If this is omitted, as is the case here, interesting features of the solution might be lost.

References

- [1] F. Soyka, O. Zvyagolskaya, C. Hertlein, L. Helden and C. Bechinger, Critical Casimir Forces in Colloidal Suspensions on Chemically Patterned Surfaces, *Phys. Rev. Lett.* 101, no. 20 (2008) 208301.
- [2] F. Schmidt, Active Matter in a Critical State: From Passive Building Blocks to Active Molecules, Engines and Droplets, PhD Thesis, Gothenburg University, December 2020, ISBN 9789180091343.
- [3] F. Schmidt, A. Callegari, A. Daddi-Moussa-Ider, B. Munkhbat, R. Verre, T. Shegai, M. Käll, H. Löwen, A. Gambassi and G. Volpe, Tunable Critical Casimir Forces Counteract Casimir–Lifshitz Attraction, *Nat. Phys.* 19, no. 2 (2023) 271–278.
- [4] F. Kümmel, B. ten Hagen, R. Wittkowski, I. Buttinoni, R. Eichhorn, G. Volpe, H. Löwen and C. Bechinger, Circular Motion of Asymmetric Self-Propelling Particles, *Phys. Rev. Lett.* 110, no. 19 (2013) 198302.
- [5] B. Küçüköz, O. V. Kotov, A. Canales, A. Y. Polyakov, A. V. Agrawal, T. J. Antosiewicz and T. O. Shegai, Quantum Trapping and Rotational Self-Alignment in Triangular Casimir Microcavities, November 2023, [arXiv:2311.17843].

Response to the Reviewers' reports

We thank all Referees for carefully reading our manuscript, as well as for their helpful comments and suggestions. We earnestly addressed them in our response and revised the manuscript accordingly. Below, we provide the point-by-point response to the Referees' comments. We refer to pages in the revised manuscript, where the changes have been made in response to the Referees suggestions, and we also provide `diff` files highlighting all changes.

RESPONSE TO REVIEWER #1

- *This is an interesting work that contributes to the manipulation of microscopic objects by precise and controllable forces and torques. As this study shows, for this purpose critical Casimir forces represent a powerful tool that can be finely tuned through the temperature of the environment and the chemical properties of the involved objects. It is very important to have a tool to self-organize ensembles of particles and to counteract stiction caused by the Casimir-Lifshitz forces.*

Here the authors explore for the first time the potential of critical Casimir torques and demonstrate that the critical Casimir torques can efficiently control the alignment of microscopic objects on nanopatterned substrates. They have managed to show experimentally and support with theoretical calculations as well as Monte Carlo simulations that circular patterns on a substrate can stabilize the position and orientation of microscopic disks. And by making the patterns elliptical the microdisks can be subject to a torque which flips them upright while simultaneously allowing for more accurate control of the microdisk position. In addition, as it is elaborated more complex patterns can also selectively trap two dimensional chiral particles and generate particle motion similar to non-equilibrium Brownian ratchets.

All the findings provide new opportunities for nanotechnological applications requiring precise positioning and orientation of microscopic objects, and as a result the publication of this study will be of high interest to several nanotechnology fields.

Therefore, I strongly suggest publication.

Our response: We are grateful to the Reviewer for the supportive comments and for strongly recommending the publication of our work.

RESPONSE TO REVIEWER #2

- *In this manuscript, Gan Wang and co-authors describe their experimental results using critical Casimir forces and torques to align and move floating nanoparticles. By changing the temperature, they can control the strength of the interaction, allowing them to turn the effect on and off. The authors explore the use of disks and spheres for trapping and orientation in- and out-of-plane. They further describe chiral microparticles and how they align. Finally, the authors describe a critical Casimir ratch, which uses a spatial variation of the hydrophobic surface to create a potential that allows for a ratcheting effect.*

Overall the manuscript is well-written, and the results are convincing.

Our response: We thank the Referee for the positive assessment of our study.

- *1) The focus of the paper is on nanoalignment, but have the authors tried to quantify the forces and torques that they are using? I see the energy potential has been determined but not the force or torque. I'd be curious to see how these numbers compare to the QED Casimir force and torque.*

Our response: In the revised manuscript, we added examples of forces and torques acting on the microdisk for the parameters of Fig. 1 of the main text (see the new Fig. S8).

In our case, the QED Casimir force is an order of magnitude weaker than the critical Casimir force (Fig. R2). Thus, we have neglected the latter in our calculations.

- *2) Fig. 2 (c,d): Eta (y-axis) should be defined/clarified. The text mentions that it is a "characteristic length," but I could not find an explicit equation. It is also not mentioned in the caption.*

Our response: We thank the Reviewer for spotting this lack of information. The y-axis of Figs. 2c and 2d is the inverse correlation length ξ of the critical fluid. We have clarified this point in the revised manuscript — see the caption of Fig. 2 on page 7, which now provides also the related equation for this characteristic length.

RESPONSE TO REVIEWER #3

- *This article describes a very interesting experiment where critical Casimir force are used to trap particles of different shapes and sizes on a patterned surface. The control parameter is the temperature difference from the critical point, which changes the Casimir interactions between the particles and between the particle and the surface. The very new and interesting result is the appearance of a torque which allows particles to stay in strange equilibrium positions. Thanks to their chirality, they can move when a temperature modulation is applied to the sample. A theoretical interpretation is also proposed. The results are interesting and new and they might have useful applications in particle assembly. For these reasons the article can be published in Nature Comm after that the following comments will be taken into account.*

Our response: We thank the Referee for positively assessing our study.

- 1) *The modulation of Casimir force to produce dynamical interactions has been already used in two references which must be quoted : Martinez et al SciPost Phys. 15, 247 (2023) and Martinez et al. Entropy 19, 77 (2017).*

Our response: We thank the Referee for pointing out these contributions, to which we now refer in the revised manuscript (page 2).

- 2) *The discussion of the figure in page 5 and 6 is not well written. Indeed it is written for disks ” we determined an effective confining potential $U_{\text{exp}}(s)$ by computing the probability $P_{\text{exp}}(s)$ ” One might think that the method is used for the disks only but I suppose that this is the case for the measure of the potential acting on the spheres. After the discussion on the disks we go back on the sphere. All of this is very confusing.*

Our response: We thank the referee for pointing out this confusing part. We addressed this issue by providing clarifications and by incorporating the calculation method of the potential earlier in the manuscript (see page 5):

In order to quantify the trapping of the particle, we used the particle trajectories to compute an effective confining potential $U_{\text{exp}}(s)$. This potential was calculated by evaluating the probability $P_{\text{exp}}(s)$ for the particle displacement from the center of the pattern ($s = 0$) and by using $U_{\text{exp}}(s) = -k_B T \ln P_{\text{exp}}(s)$.

Correspondingly, on page 6 we implemented the following amendments :

Similarly as for the microsphere, from the experimentally measured trajectories we determined effective confining potentials $P_{\text{exp}}(s)$. The main features of $P_{\text{exp}}(s)$ agree well with the theoretical calculations (see Methods “Model interaction potential”)

- 3) Page 6. The temperatures are not given both in the text and in the fig.3 caption.

Our response: We thank the Referee for spotting this typo. We added this information in the revised manuscript (see the caption of Fig. 3).

- 4) Page 10 and methods. The technique to measure theta does not seem very precise because it is based on small light differences. The accuracy must be declared. The accuracy on the position is certainly not great too because the shape of the objects changes a lot when the 3D orientation changes. It would be useful to give an error on the position accuracy.

Our response: We agree with the Reviewer that determining the precise position and orientation of the particles is challenging because of the reasons mentioned by the Reviewer. Therefore, we agree that it is relevant to provide an estimate of the errors. To address this issue, we employed the method outlined in Ref. [1] to compute the variance (σ^2) of the localization error induced by our tracking technique. The formula employed to obtain the standard deviation of the error is

$$\sigma = \sqrt{R(\Delta x_n)^2 + (2R - 1)\Delta x_n \Delta x_{n+1}},$$

where $\Delta x_1, \Delta x_2, \dots, \Delta x_n, \Delta x_{n+1}$ represent time series of single-time-lapse displacements, calculated as $\Delta x_n = x_n - x_{n-1}$. R denotes the motion blur coefficient, conventionally assigned a value of 1/6 due to the camera shutter typically remaining open throughout the entire duration of the time lapse. Based on this approach, we determined the standard deviation of the localization error for position and orientation to be 12.8 nm and 7.7°, respectively, which falls within an acceptable range compared to the particle size.

In the revised manuscript, we clarified this issue (see page 17):

In order to ensure the accuracy of the particle position and of the orientation measurements, we employed the approach outlined in Ref. [1] to calculate the variance of the localisation

error both for the position and the orientation. The resulting values are 12.8 nm and 7.7°, respectively.

- *5) The description of the method used to compute the interactions potential is too short to fully understand the technique. A section has to be added in supplementary materials with more details.*

Our response: We thank the Reviewer for this suggestion. We have now extended the Method section significantly to make our theoretical calculations clearer to a general audience (see pages 19–22). In particular, we provide more details on how we computed the interaction potentials (pages 19–20). The supplementary Sections S2 and S6 provide more details.

RESPONSE TO REVIEWER #4

- *The manuscript titled ‘Nanoalignment by critical Casimir torques’ presents a demonstration of the lateral and angular alignment of micron-size objects suspended in a solution above a patterned surface by means of the critical Casimir force. The authors perform tracking experiments to confirm that spherical, disk-shaped and non-rotationally symmetric objects can be guided by shaping an interaction potential and then switching this potential on and off by changing the temperature around the critical point. They also provide numerical computations that show that the most important aspects of the interaction can be understood on the basis of the gravitational, electrostatic and critical Casimir potentials. While these numerical results agree qualitatively with the data, interfacial interactions in liquids are complicated and the assumptions made in the present article may be too crude in some places to achieve quantitative agreement with the data. While none of the aspects shown in this paper are conceptually new, and many details, such as positional control by surface patterning [1–3], rotation by means of the critical Casimir effect [4], asymmetric structures [4], etc, have been demonstrated experimentally, it is the first time that the two-dimensional linear and rotational control of single particle trajectories by means of surface patterning and the critical Casimir force are tested explicitly. Also, the flipping of flat disks into the vertical direction is (to my knowledge) new from the experimental perspective. For this reason, the subject is timely and interesting. If the incremental step performed in this work is of sufficient interest for the readers of Nature Communications, is left to be decided by the editor. Apart from this aspect, I recommend publication after minor revision.*

Our response: We thank the Referee for carefully reading of our manuscript and for appreciating its novelty.

- *The paper is written well and clearly understandable. In some parts, such as the description of the theoretical models in Sections S2, and S4–S6, however, the paper lacks detail and many assumptions appear to be ad-hoc to the reader, especially if one is not an expert on the theory in the field of critical Casimir forces (which applies to myself). Here, more in-depth discussions on the taken (simplifying) assumptions,*

their validity, and impacts on the results would have been beneficial for understanding and clarity. Graphics are mostly well designed and facilitate a better understanding. In some points, however, they appear cluttered and pack too many details, thereby requiring the reader to study the text in all detail to understand the depicted physics.

Our response: Following the Referee’s comment, in the revised manuscript we clarified the assumptions made in the supplementary material (pages S10, S12, S31, S32, and S33).

We have also modified Fig. 3 by showing the histograms as separate panels for better readability.

- 1. *On line 50 it is stated that ‘..., the use of Casimir torques to localize, align, and manipulate the orientation of particles is yet to be established’, which may be a bit exaggerated both in light of the literature review preceding the statement and also in light of existing works, such as Ref. [4] showing the controlled rotation of asymmetric parts both in response to local heating on a surface and near a wall. Also, the rotational alignment by torques has been demonstrated for the QED Casimir effect [5].*

Our response:

We thank the Reviewer for this comment. We agree that the statement in the previous version of the manuscript was too strong, and we have therefore rewritten it to focus it on critical Casimir torques, and not Casimir torques in general (page 3).

However, the use of critical Casimir torques to localise, align, and manipulate the orientation of particles is still to be established.

- 2. *Plotting in Figure 2c the phase diagram with coordinate R/ξ instead of ξ/R appears a bit counter-intuitive. Using the inverse would allow the reader to see to directly see the relation of Temperature, particle size and correlation length.*

Our response: Presenting results in terms of R/ξ is customary in the critical Casimir field. This choice is motivated by the scaling behavior of the correlation length ξ , which follows the power law $\xi \sim (T - T_c)^{-\nu}$ as the temperature T approaches the critical temperature T_c . Since $\nu > 0$, the so computed scaling variable is directly proportional to the temperature difference $\Delta T = T - T_c$, resulting in both axes showing a proportional increase in their values. For the Reviewer’s reference, in Fig. R1, we

Figure R1. Panels (c) and (d) of Fig. 2.

replicate Fig. 2 of the main text using ξ/R instead of R/ξ , which shows an inverse behavior of the left and right axes. Although this is still a valid representation, we prefer to adhere to the commonly used variable R/ξ .

- 3. Line 180: Should the second width not read σ_y instead of σ_x ?

Our response: We thank the Referee for spotting this typo, which we have corrected.

- 4. Figure 3: The histograms along the coordinates of the trajectory plot in a and c are so small and without proper scale that they are hard to read and therefore give little information other than the qualitative distribution (which shows that the integration time was probably too short to fully probe the distribution). Also the ‘short black vertical segment’ mentioned in the caption is very hard to identify. Tablets a and c could be made larger by squeezing the space between the figures. In the caption, you mention that the orientation is ‘sharply confined between -7° and $+7^\circ$ ’. Does this refer to σ_θ (i.e. the width of a Gaussian distribution), is it the FWHM, or even the min/max of the distribution? A bit more quantitative rigor would be appreciated here.

Our response: We modified this figure (Fig. 3) to show histograms as separate panels for clarity, so that the short black vertical segment can also be better identified. The range -7° to 7° refers to the angles θ read off from panel (d). In the revised manuscript,

we also provide the Gaussian widths from the fit (see the caption to Fig. 3).

- 5. *Without reading the supplementary information (and even then) it is very difficult to understand how you modeled and computed the potentials (see comments below).*

Our response: We thank the Reviewer for pointing this out. We agree that the description of the model potential was insufficiently detailed in the initial version of the manuscript. In response to this feedback, we have extensively expanded the theoretical part of the Methods section, particularly focusing on providing more comprehensive explanations of how we computed the interaction potentials (see pages 19–20 in the revised manuscript). We hope that these revisions have enhanced the clarity of our theoretical methods used for the manuscript.

- *Remarks on the supplementary information*

1. *Equation S5 is the basic sum of potentials. Yet, at distances around 100 nm, despite not being the dominating force, the quantum mechanical Casimir effect should be taken into account, as demonstrated in the literature [3]. On line 366 (methods section of the main article) you note that you follow Hertlein et al (your ref. [19]) to neglect dispersion forces. However, Hertlein et al used a polystyrene particle above a silica surface, which are both dielectric materials leading to strongly reduced QED Casimir forces, while in your case you have SiO₂-Au interactions. Depending on the degree of oxidation of the SiO₂, the original conductivity of the oxidized Si, and the dielectric function of the water-2.6 lutidine mixture, the resulting Casimir forces at ≈ 120 nm shortest separations (judging from Figure S7) may be significantly higher than in the experiment of Hertlein et al. It may not reach the strength of the one measured by Schmidt et al [3] who used functionalized gold against a gold flake but also I expect the force in your case not to be lower by an order of magnitude than for Au-Au. Therefore, at least a rough estimate of the expected Casimir potential in relation to the other terms in Equation S5 should be given. This requires knowledge of the dielectric function of the suspension. The respective functions of glass and Au can be taken from the literature.*

Our response:

We thank the Reviewer also for this comment. We estimated the QED Casimir potential (dispersion force) for a disk above a gold-coated and an uncoated substrate

Figure R2. **Critical Casimir vs QED Casimir potentials.** Critical Casimir and QED Casimir potentials between a disk and a substrate expressed in units of $k_B T$ as functions of the distance from the substrate for (a) gold-coated and (b) uncoated (glass) substrate. The microdisk is oriented parallel to the substrate.

following the approach outlined in Ref. [2]. The results are presented in Fig. R2, where we compare these potentials with the corresponding critical Casimir potentials. The calculations presented in this figure confirm that the QED Casimir potential is about two orders of magnitude weaker than the critical Casimir potential (with both potentials taken to be zero at macroscopically large separation distances). We have therefore opted to exclude the QED contribution from our models. In the revised manuscript, we provide a new section (Section S2C) and a new Fig. S7 showing the comparison of the QED Casimir and the critical Casimir potentials.

- 2. While the use of the Derjaguin approximation should give quantitatively good results from small α , I would expect significant deviations for larger α , γ . This, in addition to the very crude approximation of a homogeneous charge distribution on

both the substrate and the microdisk leading to Equation S9, may explain the rather large quantitative differences between the numerical and experimental results in Figure 2 (and Figure 5) of the article. It would be rather simple to replace Equation S9 by an empiric relation or interpolation derived from electrostatic simulations in an appropriate tool (e.g. Comsol or Ansys) taking into account the feedback between $l(x, y)$, U_e (or ue ?) and $\sigma(x, y)$ for different α . While this is a nice-to-have feature for the present article in its present state, it would (in my opinion) improve the quantitative understanding of your experiment significantly.

Our response: While we acknowledge that the Derjaguin approximation may not yield precise quantitative results for tilted disk configurations, it still offers valuable qualitative insights and effectively captures the underlying physics related to disk flipping (Fig. 2) and ratchet motion (Fig. 5). Although there are noticeable quantitative discrepancies, we consider these differences to be relatively minor and emphasize that our primary goal is to achieve a qualitative understanding of the observed rich phenomena rather than precise quantitative agreement.

We appreciate the Referee’s suggestion regarding the potential replacement of Eq. (S9) with an empirical relation. However, this approach is not straightforward due to several factors, particularly that the empirical relation is not *a priori* known. If we understand the Referee’s suggestion correctly, it proposes to solve a ‘reverse problem’ by aligning the theoretical and experimental data and presuming more intricate electrostatic interactions. They can only be solved numerically, with additionally considering surface inhomogeneity. Besides being computationally demanding, this approach introduces extra fitting parameters, and it remains uncertain whether fitting can be conclusively carried out in this scenario. Even when considering only two fitting parameters, we observe a range of surface charges and Debye lengths which yield qualitatively similar results for the configuration probability. In this regard, we emphasize that our primary objective with these theoretical calculations is to qualitatively understand the physical mechanisms underlying the experimental observations. We expect that our current calculations effectively capture these phenomena despite certain quantitative discrepancies.

- 3. Similarly, the assumption that the disk is perfectly parallel to the substrate on line

284 as well as the neglect of 3-body effects in the critical Casimir and electrostatic interactions may imperil quantitative reliability of the numerical results for the ratchets. However, I understand that such computations may be beyond the scope of the present article and given that I am not an expert on the theory you apply, this comment may rather be seen as a curious question.

Our response: As the Reviewer correctly points out, we assumed that the disk is parallel to the substrate. However, we believe this approximations should not significantly influence the interaction potential as the disk tilt is small in most situations we consider experimentally. We added a comment in the Method section (page 21):

Following experimental observations, we assumed that the microdisk is always in the parallel configuration. For simplicity of the calculations, we ignored disk tilting and assumed that the disk is parallel to the substrate. Taking into account that tilt angles in the parallel configuration are small, this assumption influences the interaction potential only slightly.

We neglected three-body interactions because we have only two objects (a disk and a substrate), while the electrostatic interactions are additive in any case.

- 4. *The motivation for the pre-factor of the weight in Equation S10 remains elusive. Due to the central role of this weight, a little discussion would be in order.*

Our response:

Within MC simulations, in order to preserve detailed balance, the weight assigned to each configuration must be proportional to the probability \mathcal{P} of the occurrence of that configuration. In general, $\mathcal{P} \sim \exp[-\mathcal{H}/(k_{\text{B}}T)]$, where \mathcal{H} is a Hamiltonian, which depends on six configurational variables and six generalised momenta. The weight factor in Eq. (S15) follows from integrating out all degrees of freedom in \mathcal{P} except for Δ , Σ , α , and γ . In the revised manuscript, we provide some details of these calculations (see the new Section S4 A).

- 5. *Distributions for the starting parameters in Monte Carlo simulations are essential for the outcome. Why are uniform distributions for all parameters rectified here? At least for the initial distance to the surface, I would have expected some peaked distribution at the equilibrium distance.*

Our response: The use of peaked distributions could enhance efficiency of simulations. However, as these distributions are not readily available to us, we opted for uniform distributions within reasonable ranges. Simulation results should remain robust and not significantly dependent on initial distributions for a sufficiently large number of simulation steps.

The necessary number of steps has been determined by monitoring the average particle energy. We have checked that for all considered values of parameters describing the system, 2000 Monte Carlo steps is enough for the average energy as a function of step number to not show any systematic trend. We added a comment concerning this issue in the supplementary material (page S28).

- 6. *The separated single lines visible in Figure S8 (especially b and d) show that the final distribution is probed with insufficient resolution. While the qualitative outcome is clear and also analyzed correctly in the text, the simulation should be repeated until the distribution is sufficiently smooth. If this is omitted, as is the case here, interesting features of the solution might be lost.*

Our response: The single lines visible in Fig. S10 do not indicate errors but rather orientations with low but non-zero probabilities. Near a transition point, these probabilities are higher, leading to larger numbers of both parallel and perpendicular disk orientations, as evident in panel (c). As we move away from the transition, the number of parallel (panel (b)) and perpendicular (panel (d)) orientations decreases but remains non-zero.

For the scenario depicted in this figure, we conducted additional simulations and calculated the standard deviations for the corresponding probability functions, which are now presented on each panel of Fig. S10.

-
- [1] C. L. Vestergaard, P. C. Blainey, and H. Flyvbjerg, Optimal estimation of diffusion coefficients from single-particle trajectories, *Phys. Rev. E* **89**, 022726 (2014).
- [2] F. Schmidt, A. Callegari, A. Daddi-Moussa-Ider, B. Munkhbat, R. Verre, T. Shegai, M. Käll, H. Löwen, A. Gambassi, and G. Volpe, Tunable critical Casimir forces counteract Casimir–Lifshitz attraction, *Nat. Phys.* **19**, 271 (2023).

REVIEWERS' COMMENTS

Reviewer #2 (Remarks to the Author):

The authors have done an excellent job of addressing questions and comments. In my opinion the manuscript should be published in Nature Comm.

Reviewer #3 (Remarks to the Author):

The authors took care of accurately answering to the referees comment. I have a comment on the critical Casimir versus QED Casimir. I agree with the authors that this is a very negligible effect because of the dielectric constant of the water. The effect of the dielectric on QED Casimir was precisely measured in PRB 98, 201408(R) (2018) that the authors could quote to strength their claim.

After that the article can be published.

Referee report

I would like to thank the authors for their comprehensive answers which clarified most points. The revised version of the manuscript presents the work more clearly and does not lack any significant information. I recommend publication after correcting very minor errors that have been introduced in the new version. From my side, no further review round is required.

Remarks main text

1. Figure 3: Indicators for b, c, d, f, g, h are missing. The thick white line in the small micrographs supposedly represents the $2\ \mu\text{m}$ scale bar, as in the larger image, but it lacks an indicator or description in the text.
2. In response to your answer to my previous comment about the estimation of the quantum Casimir force (or potential), I would like to leave a comment for the authors that does not need to influence the current paper.
The Derjaguin approximation produces an error proportional to a/R for the energy (see for example [1]), where a is the object distance, and R is the surface radius. As the disks in your case supposedly have very small surface edge radii, and the location of these edges are at the closest separation, the error might be significant. There is an easy way (for the future) to significantly reduce such uncertainties by using modern 3D computation codes, such as `scuff-em` that provide a very easy-to-use interface and produce rather accurate results. In my opinion, despite being discussed to be of sub-leading order throughout the literature of the critical Casimir effect, the quantum Casimir effect should be considered in situations where (unstable) equilibria of objects with non-smooth surfaces or aspect ratios not deviating more than a factor ~ 10 from unity (such as the trapping of your disks), are investigated. I do not insist on this point to be taken into account in the present paper, as you clearly state that you are interested in the *qualitative* behavior. However, if any quantitative agreement is desired, the quantum Casimir force amounting to a few percent up to order 1 relative to the critical Casimir force – depending on the coatings and the distance regime – needs to be considered in a more rigorous way, including the very dielectric spectra of all involved materials, and a proper (i.e. not Derjaguin) account of the geometry. The same is true for electrostatic interactions. It is well known that charges are *not* evenly distributed (even on flat metal surfaces), and that this creates a significant background in any force-sensitive measurement at sub-micron separations. Note further that the use and formalism of Hamaker coefficients is deprecated. Instead, use of the Lifshitz theory is encouraged for simple estimations, while more precise codes should be used for any quantitative statement on the behavior of real geometries involving edges and finite object size. Again, see [1].

Remarks on the supplementary information

1. Equation S11. The first term is missing a factor $1/h^2$.

2. Caption of Figure S7: The caption text is cut off. (This might just be a formatting problem.)

References

- [1] M. Bordag, G. L. Klimchitskaya, U. Mohideen and V. M. Mostepanenko, *Advances in the Casimir Effect*, Oxford University Press, October 2014, ISBN 978-0-19-871998-4.

Response to Reviewers' Comments

Reviewer #2 (Remarks to the Author):

The authors have done an excellent job of addressing questions and comments. In my opinion the manuscript should be published in Nature Comm.

We thank the reviewer.

Reviewer #3 (Remarks to the Author):

The authors took care of accurately answering to the referees comment. I have a comment on the critical Casimir versus QED Casimir. I agree with the authors that this is a very negligible effect because of the dielectric constant of the water. The effect of the dielectric on QED Casimir was precisely measured in PRB 98, 201408(R) (2018) that the authors could quote to strength their claim.

After that the article can be published.

We have now addressed this suggestion by the reviewer and added this reference.

Reviewer #4 (Remarks to the Author):

I would like to thank the authors for their comprehensive answers which clarified most points. The revised version of the manuscript presents the work more clearly and does not lack any significant information. I recommend publication after correcting very minor errors that have been introduced in the new version. From my side, no further review round is required.

We thank the reviewer. We have now corrected the remaining errors (see below).

Remarks main text

1. Figure 3: Indicators for b, c, d, f, g, h are missing. The thick white line in the small micrographs supposedly represents the 2 μm scale bar, as in the larger image, but it lacks an indicator or description in the text.

We thank the reviewer. We have added the letters and added a clarification in the caption about the error bar.

2. In response to your answer to my previous comment about the estimation of the quantum Casimir force (or potential), I would like to leave a comment for the authors that does not need to influence the current paper.

The Derjaguin approximation produces an error proportional to a/R for the energy (see for example [1]), where a is the object distance, and R is the surface radius. As the disks in your case supposedly have very small surface edge radii, and the location of these edges are at the closest separation, the error might be significant. There is an easy way (for the future) to significantly reduce such uncertainties by using modern 3D

computation codes, such as scuff-em that provide a very easy-to-use interface and produce rather accurate results. In my opinion, despite being discussed to be of sub-leading order throughout the literature of the critical Casimir effect, the quantum Casimir effect should be considered in situations where (unstable) equilibria of objects with non-smooth surfaces or aspect ratios not deviating more than a factor ~ 10 from unity (such as the trapping of your disks), are investigated. I do not insist on this point to be taken into account in the present paper, as you clearly state that you are interested in the qualitative behavior. However, if any quantitative agreement is desired, the quantum Casimir force amounting to a few percent up to order 1 relative to the critical Casimir force – depending on the coatings and the distance regime – needs to be considered in a more rigorous way, including the very dielectric spectra of all involved materials, and a proper (i.e. not Derjaguin) account of the geometry. The same is true for electrostatic interactions. It is well known that charges are not evenly distributed (even on flat metal surfaces), and that this creates a significant background in any force-sensitive measurement at sub-micron separations. Note further that the use and formalism of Hamaker coefficients is deprecated. Instead, use of the Lifshitz theory is encouraged for simple estimations, while more precise codes should be used for any quantitative statement on the behavior of real geometries involving edges and finite object size. Again, see [1].

References

[1] M. Bordag, G. L. Klimchitskaya, U. Mohideen and V. M. Mostepanenko, Advances in the Casimir Effect, Oxford University Press, October 2014, ISBN 978-0-19-871998-4.

We thank the reviewer for this clarification, which will indeed be very useful for future works.

Remarks on the supplementary information

1. Equation S11. The first term is missing a factor $1/h^2$.

We have now corrected it.

2. Caption of Figure S7: The caption text is cut off. (This might just be a formatting problem.)

We have now resolved this formatting issue.